# Evaluation of the Microbial Quality of *Hermetia illucens* Larvae for Animal Feed and Human Consumption: Study of Different Type of Rearing Substrates

**DOI:** 10.3390/foods13101587

**Published:** 2024-05-20

**Authors:** Lenaïg Brulé, Boris Misery, Guillaume Baudouin, Xin Yan, Côme Guidou, Christophe Trespeuch, Camille Foltyn, Valérie Anthoine, Nicolas Moriceau, Michel Federighi, Géraldine Boué

**Affiliations:** 1Oniris, Institut National de Recherche Pour l’Agriculture, l’Alimentation et l’Environnement (INRAE), SECurité des ALIments et Microbiologie (SECALIM), 44300 Nantes, France; lenaig.brule@inrae.fr (L.B.); boris.misery@oniris-nantes.fr (B.M.); xin.yan@oniris-nantes.fr (X.Y.); camillefoltyn59@hotmail.fr (C.F.); valerie.anthoine@oniris-nantes.fr (V.A.); nicolas.moriceau@inrae.fr (N.M.); 2Cycle Farms, 6 Boulevard des Entrepreneurs, 49250 Beaufort en Anjou, France; guillaume.baudouin@cyclefarms.com; 3MUTATEC—1998, Chemin du Mitan, 84300 Cavaillon, France; c.guidou@mutatec.com (C.G.); c.trespeuch@mutatec.com (C.T.); 4EnvA/Anses, Laboratoire de Sécurité des Aliments, 94700 Maisons-Alfort, France; michel.federighi@vet-alfort.fr

**Keywords:** edible insect, black soldier fly, BSF, food safety, microbiological risk assessment, insect farming, rearing substrate

## Abstract

In the context of climate change and depletion of natural resources, meeting the growing demand for animal feed and human food through sufficient, nutritious, safe, and affordable sources of protein is becoming a priority. The use of *Hermetia illucens*, the black soldier fly (BSF), has emerged as a strategy to enhance the circularity of the agri-food chain, but its microbiological safety remains a concern. The aim of the present study was to systematically review available data on the microbiological quality of BSF and to investigate the impact of using four different rearing substrates including classic options allowed by the EU regulation (cereals, fruits, vegetables) and options not allowed by EU regulations regarding vegetable agri-food (co-products, food at shelf life, and meat). A total of 13 studies were collected and synthesized, including 910 sample results, while 102 new sample results were collected from the present experiments in three farms. Both datasets combined revealed a high level of contamination of larvae, potentially transmitted through the substrate. The main pathogenic bacteria identified were *Bacillus cereus*, *Clostridium perfringens*, *Cronobacter* spp., *Escherichia coli*, *Salmonella* spp., and *Staphylococcus aureus* coagulase-positive, while *Campylobacter* spp. and *Listeria monocytogenes* were not detected. Any of these four substrates were excluded for their use in insect rearing; however, safety concerns were confirmed and must be managed by the operators of the sector using microbial inactivation treatment after the harvest of the larvae in order to propose safe products for the market. The results obtained will guide the definition of the control criteria and optimize the following manufacturing steps.

## 1. Introduction

The overall consumption of proteins, including those of animal origin, has increased in recent decades and is expected to expand significantly until 2050 [1,2,3]. The need to ensure global food security represents a significant challenge over the next three decades. The current model of animal protein production, relying on cattle, pigs, poultry, and fish, has demonstrated various limitations, leading to the emergence of the need for a protein transition. To date, traditional agricultural practices face limitations in terms of key resources such as arable land, energy, and water, impeding their ability to sufficiently produce quality protein. Moreover, the current food production system is considered unsustainable due to its adverse environmental impacts, including greenhouse gas emissions (GHG) and soil depletion. The development of a safe and sustainable food system becomes even more imperative considering the projected expansion of the global population. In this context, insects have emerged as a promising alternative to traditional animal production due to their ease of rearing, their better bioconversion rate, and their low environmental impact [4,5]. Edible insects can be used as feed and food, and are highly nutritious, rich in proteins, healthy fats, vitamin, and minerals [6]. Moreover, insects offers versatility in their applications producing oil, chitin, fertilizer, and other derived products [7,8,9,10,11].

Among the available insect species, *Hermetia illucens*, commonly known as the black soldier fly (BSF), is recognized for its applications in animal feed production and bio-waste management [12]. The larvae of *Hermetia illucens* and their associated specific compounds (protein, fat, and chitin) exhibit potential for diverse applications including aquaculture [13,14], livestock feed [15], human food [16], as well as other promising areas like biofuel production [17] and the development of bioactive coatings [18]. Bioconversion capabilities of BSF larvae can address the significant challenge of food waste, which reaches up to 10 million tons per year in France [19]. In addition, *Hermetia illucens* larvae are highly nutritious and are often used for animal feed [20]. The BSF offers the potential to address the environmental detriments of food waste and meet the rising protein demand by converting food waste into biomass as alternative food sources. Rearing farms can benefit from low-cost substrates and efficient waste management, whereas they must comply with regulatory requirements.

Globally, in industrialized countries, edible insects for human consumption have to respect the same standard sets as other foods. In EU, edible insects are considered as “products of animal origin” (Regulation (EC) 853/2004) [21]. Moreover, according to the Regulation (EC) 2015/2283 [22] of the European Parliament and of the Council on novel foods, insects and their products are considered as novel foods and have to be authorized before being placed on the food market, only after a safety assessment by the European Food Safety Authority (EFSA). In addition, a draft regulation amending Annex III of Regulation No. 853/2004 was published for public comment in January 2019 (Ares (2019) 382900), which regards specific hygiene requirements for insects intended for human consumption. This draft regulation does not contain new provisions specifically for insects, but rather reiterates rules from other legislations that are already applicable [23,24,25]. For example, insects can only be reared on substrates of plant origin or certain specifically permitted substrates of animal origin, such as fishmeal and hydrolyzed protein from non-ruminants (Article 4, also under Regulation (EC) No 1069/2009 [26] and No 142/2011 [27]); substrates used for feeding insects must not contain manure, catering waste, or other waste (Article 5, also under Regulation (EC) No 1069/2009). Several substrates for rearing *Hermetia illucens* have been tested, including non-authorized substrates, like pig manure [20] or organic (vegetables, foods, and brewery) wastes [28,29,30,31]. On the other hand, adult BSFs prefer millet porridge mash substrate for oviposition [20].

Moreover, research has suggested that the gut microbial community in BSFs may be greatly influenced by different feeding substrates, such as cooked rice, calf forage, and food/vegetable wastes [32,33,34,35,36,37]. A study from Wynants et al. [32] demonstrates the variation in the microbiota of BSF larvae raised in diverse facilities, each employing unique rearing methods. Thus, the substrates provided for insect rearing emerge as a critical factor in ensuring food/feed safety [16,37]. It is notable that the digestive tract of insects is not removed regardless of the form in which they are consumed, and the fasting phase applied before slaughter does not significantly affect the digestive content. In addition, insects grow in a substrate that they consume and subsequently deposit, known as frass, which poses a challenge for separation and can act as a medium for microbiological hazards. Harvesting insects via sieving does not ensure complete separation of larvae and frass, leaving the finest particles adhering to the larvae. As a result, food pathogens can be present both in the substrates and intestinal tract of insects [16,38,39], justifying the need to put in place a processing step with thermic treatment and the implementation of good manufacturing practices as well as internal controls. Numerous publications dedicated to edible insects focus on their microbiological quality [40,41,42]. It is generally accepted that entomopathogenic microorganisms do not belong to the same phyla as the pathogens relevant to food safety. It is also acknowledged that insects can serve as vectors of these hazards throughout the breeding and processing cycles. However, to date, little attention has been paid to the assessment of the impact of substrates on the microbiological quality of *Hermetia illucens* larvae for both animal feed and human food at the stage of unprocessed raw materials.

The aim of the present study is to evaluate the microbiological quality of black soldier fly larvae, performing a systematic review of the literature and an experimentation conducted in three farms using four different rearing substrates including traditional recipes (cereals, fruits, vegetables), vegetable agri-food co-products, former foodstuffs arriving at shelf life, and meat.

## 2. Materials and Methods

### 2.1. Systematic Review of the Microbiological Quality of BSF Larvae, Substrate and Frass

Systematic research was performed to identify the main pathogenic bacteria and collect their levels of contamination in black soldier fly samples, including raw and processed larvae as well as their substrate and frass. 

#### 2.1.1. Search Strategy

The Preferred Reporting Items for Systematic Reviews and Meta-Analyses (PRISMA) guidelines for systematic reviews were used [43]. Two electronic bibliographic databases were used, namely PubMed and ScienceDirect, using the following search criteria: PubMed: ((“Black Soldier Fly” OR “*Hermetia illucens*”) AND (“microb*” OR “cultural method*” OR “sequenc*” OR “metagen*” OR “metabarcoding” OR “bacteri*” OR “profil*” OR “dynamic*”), and Web of Science with two searches (Title, abstract, keywords: (“*Hermetia illucens*” OR “Black soldier fly”) AND (“microbiol” OR “Microbiological” OR “microbiol community” OR “microbiota” OR “sequencing” OR “metagenonic” OR “metabarcoding”)) and (Title, abstract, keywords: (“*Hermetia illucens*” OR “Black soldier fly”) AND (“bacteri*” OR “profil*” OR “dynamic”)). The last search was performed on 27 September 2023 with no limitation on year or language. Duplicates were removed on Zotero. Additionally, the list of references of selected articles was analyzed to identify further articles.

#### 2.1.2. Eligibility Criteria

Articles providing data with cultural methods on research and/or quantification of main pathogenic bacteria found in BSF larvae (raw or processed) as well as substrate and frass were eligible for data extraction with no restriction on the developmental stage of the larvae. The main reasons for exclusion were related to experimentations of different BSF feed, omics data, applications in waste conversion, bacteria strain isolation, and antibiotic-resistance-related articles. Titles and abstracts were shared and screened by three reviewers, with discussions to resolve disagreements when necessary.

#### 2.1.3. Extraction, Collation, and Standardization of Data

Creating the list of references and screening articles were performed using Zotero (6.0.30). Data from each article were collected and synthesized in an Excel table, including the year, the country of origin, specific conditions, the process tested, the number of samples, the methods used, and results. Levels of microbiological hazards were reported in log cfu/g.

### 2.2. Experimental Design and Sample Collection

To assess microbial contamination during the rearing of BSF larvae, three industrial rearing facilities located in France, further referred to as company F1, F2, and F3, were involved in this study (Figure 1 and Table 1). For each company, 1 to 3 consecutive rearing cycles were examined. At each sampling point, three technical replicates of larvae, native substrate, and frass at harvest were collected.

For all farms, 250 g of substrate, frass, and BSF larvae were taken for each of the 1 to 3 cycles and at each sampling point, yielding a total of 10 to 48 samples per company. In total, 103 samples were collected from the three industrial locations. Samples were immediately stored in a freezer (−20 °C) until further transport and then stored at −20 °C in the laboratory until analysis.

### 2.3. Rearing Condition

Depending on the rearing condition of each company, fifth-instar larvae were produced on different diets (Table 1) and harvested for sampling. The selection of substrates aimed to include both the traditional substrate commonly used in *Hermetia illucens* rearing, as well as alternative substrates aligned with waste reduction strategies. The traditional substrate chosen was the one typically used by each participating farm. The alternatives included by-products from the agri-food industry, products at expiration date from supermarkets, and grounded beef during their shelf life. These substrates were chosen to reflect a variety of potential feed sources that could influence the contamination of *H. illucens* larvae differently, aligning with the objectives of our study to explore sustainable and diverse feeding options for larval development.

At Farm 2, the rearing conditions from third to fifth (harvest stage) instar were in a plastic container (60 × 60 × 12 cm) containing 16 kg of substrate within a climatic chamber at 27 ± 0.5 °C and 30 ± 1% of relative humidity (RH). Each container was inoculated with 10,000 third-instar larvae. After 7 days from when the larvae reached the fifth larval instar, we harvested them. At Farm 3, the rearing conditions were 27 ± 2 °C; 65 ± 2% RH. At Farm 1, the larvae were fattened at 6 or 7 days after hatching within a climatic chamber at 26 to 29 °C and 75 to 80% RH and harvested at 14 days after hatching, or 7 days after being placed in tanks. For further information, the substrate used as “organic poultry food” by Farm 3 is a blend of wheat, corn, soybean meal, calcium carbonate, barley, wheat bran, corn, corn gluten meal, sunflower seed meal, alfalfa, dicalcium phosphate, and paprika extract (Table 1). 

### 2.4. Sample Preparation and Microbiological Analysis

After one night of defrosting at 4 °C, only the larvae underwent a grinding step of 10 s three times (Blender Nutriboost 23180-56, Russell Hobbs, Courbevoie, France). The mesophilic, psychrotrophic, and endospore aerobic bacteria were quantified with a fraction of 25 g of sample diluted to the tenth with buffered peptone water (VWR) in stomacher plastic bag. The samples were stomached at 25.5 stroke/s for 2 min with a lab paddle blender (Masticator^®^, IUL, Barcelona, Spain).

After a phase of revival of 1 h at room temperature, serial dilutions were performed in buffered peptone water to 10^−8^. The total mesophilic flora and bacterial endospore counts were determined with the pour-plate method on Plate Count Agar (PCA, Biokar diagnostics) or PCA with starch 2 g/L at 30 °C for 72h after a heat treatment at 80 °C for 10 min. The spread plate technique was used to count the psychrotrophic bacteria on PCA at 6.5 °C for 10 days. The major food pathogens were investigated and the enterobacteria, lactic bacteria, and yeasts/molds were quantified according to the standards used in the LAB Eurofins (Table 2).

### 2.5. Statistical Analyses

Statistical analyses were conducted using Palissade StatTools (Version 7.6.1). Kruskal–Wallis tests were used to assess differences in microbial load among rearing substrate types (food at shelf life, co-product, classic, meat-based products), sample types (larvae, substrate, frass), and across the three farms. The χ^2^ test (*p*-value < 0.05) was performed to evaluate the normality of the distribution. Means were compared using Kruskal–Wallis followed by Bonferroni-corrected Mann–Whitney post hoc test. Statistical significance was determine-d at a threshold of *p*-value < 0.05.

## 3. Results

### 3.1. Data Extracted from Systematic Review

#### 3.1.1. Synthesis of the Systematic Search

A total of 680 articles were collected following our systematic research protocol (Figure 2). After duplicates were removed, the titles and abstracts for 564 articles were screened following eligibility criteria, resulting in 24 articles being pre-selected. After full-text analysis, 13 studies were selected for data extraction (Appendix A). They included different types of samples as summarized in Table 3 with raw and processed larvae (asphyxia, boiled, desiccated, dried, fat extracted, frozen slaughter, high pressure, oven-dried, powder, solar-dried, and toasted), substrate, and frass. The different levels of bacterial indicators and main pathogenic bacteria are reported in Table 4.

#### 3.1.2. Summary of Articles Included in the Systematic Review on *Hermetia illucens*

Among the 13 eligible articles, the study of raw larvae is predominant (n = 10) compared to processed larvae (n = 7), substrates (n = 5), or frass (n = 6) (Table 3). In five articles the slaughter method was not specified, most of the raw forms of larvae were slaughtered by freezing (n = 5). Other studies analyzed heat-treated larvae, boiled (n = 4), dried (n = 2), high-pressure processed (n = 3), solar-dried (n = 1), or toasted (n = 1). 

#### 3.1.3. Microbiological Contamination of *Hermetia illucens* Larvae, Substrates, and Frass 

Table 4 presents a synthesis of the microbiological contamination of analyzed data, representing the ranges found in the literature. The detailed levels for all samples and conditions are reported in Appendix A. For each experiment, the number of samples analyzed was low, ranging from 1 to 9 or unspecified. Only one study provided value for *Hermetia illucens* fat. 

The contamination levels found in raw larvae were generally higher than that in heat-treated or dried larvae (>10.7; 8.1; 7.8 cfu/g, respectively). *Listeria monocytogenes* was not detected in any insect samples tested. The *Aspergillus* spp. indicator was searched only once in the survey data.

### 3.2. Experimental Results

#### 3.2.1. Microbiological Analysis for Bacterial Indicators

Microbiological results obtained for bacterial indicators are summarized in Table 5, Table 6 and Table 7 per rearing farm, per type of sample, and per category of substrate, respectively. They are also reported in Figure 3 for better representation of the variability between samples.

For all substrates investigated, the general trend observed was higher levels of contamination in larvae and frass than their substrates in terms of total mesophilic aerobic bacteria, aerobic endospore bacteria, and *Enterobacteriaceae* (Figure 3). More particularly, in larvae, mesophilic aerobic bacteria and aerobic endospore bacteria ranged, from 6.9 to 9.2 log cfu/g and from 6.1 to 8.7 log cfu/g, respectively, and in frass samples from 7.8 to 10.2 log cfu/g and from 6.1 to 8.7 log cfu/g (Figure 2), respectively. These indicators are lower in substrates 3.6 to 8.1 log cfu/g for mesophilic aerobic bacteria and 3.2 to 6.9 log cfu/g for aerobic endospore bacteria. Regarding lactic acid bacteria, a large variability was observed in substrates (from < 1 to 7.5 log cfu/g) and fewer variables with high contamination in larvae (4.7 to 7.5 log cfu/g) and frass (5.2 to 7.3 log cfu/g). For the other indicators including yeast, molds, and psychrotrophic aerobic bacteria, there is no general trend between the three types of samples.

A Kruskal–Wallis test was used to analyze differences between three farms according to their differences in microbial load of total mesophilic flora, endospores, lactic acid bacteria, Enterobacteriaceae, psychotrophic bacteria, and yeasts/molds. No significant difference in microbial load was found between farms except for the molds, psychotrophic bacteria, and bacterial endospores. Bonferroni’s corrected Mann–Whitney post hoc tests revealed significant differences in contamination levels of molds, psychrotrophic bacteria, and bacterial endospores between Farm 1 and Farm 3. For a better understanding of the impact of substrate on the microbial quality of larvae, a Kruskal–Wallis test was used to assess significant differences in contamination levels of microorganisms based on rearing substrate types (food at shelf life, co-product, classic, meat-based product). No significant difference in microbial load was found between the rearing substrate types except for bacterial endospores, molds, and psychotrophic bacteria. For vegetable co-products, the bacterial endospore load was 5.8 ± 1.1 log cfu/g, whereas it was 7.4 ± 1.5 log cfu/g for meat-based products. Mold levels varied from 1.5 ± 0.5 log cfu/g (food at shelf life) to 3.5 ± 2 log cfu/g (meat-based product). For food at shelf life, the psychotrophic bacteria load averaged 3.08 ± 0.8 log cfu/g, whereas it was 5.3 ± 1.6 log cfu/g for meat-based products. The same non-parametric analysis was performed to compare microbial load according to the type of samples (frass, larvae, and substrate). A significantly higher contamination in enterobacteriaceae, bacterial endospores, lactic acid bacteria, and molds was observed in the frass compared to the substrate (*p*-value < 0.05). The contamination levels found in the frass are the same as those found in the larvae except for psychotrophic bacteria, total mesophilic bacteria, and yeasts. Significant differences in psychotrophic bacteria, total mesophilic bacteria, and yeasts were observed between the three types of samples.

#### 3.2.2. Detection and Quantification of Major Bacterial Pathogens in Samples Analyzed

Results of detection of main pathogenic bacteria are synthesized in Table 8 for each of the three farms included in the experimental plan. Table 9 provides results of the different types of substrates used, including traditional recipe (cereals, fruits, vegetables), vegetable agri-food co-products, food at shelf life, and meat. 

All samples analyzed from the three farms (substrate, larvae, and frass) were free of *Campylobacter* spp., coagulase-positive *Staphylococci*, and *Listeria monocytogenes* (Table 8). Only one frass sample (Farm 3) was found to be positive for *Salmonella* spp. 

Some pathogens like *Cronobacter* spp. and *Escherichia coli* β-glucuronidase-positive seem to be ubiquitous as they were found in substrate, larvae, and frass. *Clostridium perfringens* presumed was mainly found in larvae, unlike *Bacillus cereus*, which was only found in substrate and frass, and absent from larvae (Appendix A). Substrates used in Farm 2 were only contaminated with *Bacillus cereus* (presumptive), while the two other farms were also positive for *Clostridium perfringens*, *Cronobacter* spp., and *E. coli*. However, in Farm 2, more frass samples were positive for *Bacillus cereus* and *Clostridium perfringens* than in Farms 1 and 3. Of all the substrates tested, vegetables showed a significant difference, with a higher percentage of contaminated substrate, larvae, and frass samples. This can be explained by the fact that vegetables are in contact with their environment, particularly the soil (Table 9).

## 4. Discussion

The present study analyzed the microbiological quality of *Hermetia illucens* reared on different substrates for their introduction in animal feed and human food through a systematic review of the available data and an experimental plan performed in three different farms. The samples analyzed larvae and their associated rearing substrates and frass in three different farms using substrates based on traditional recipes, agri-food co-products and former foodstuffs, leading to a large number of samples (n = 103). The criteria included the main pathogenic bacteria found in feed and food (*Bacillus cereus group*, *Campylobacter* spp., *Clostridium perfringens*, *Cronobacter* spp., *Escherichia coli*, *Listeria monocytogenes*, *Salmonella* spp., *Staphylococcus aureus* (coagulase-positive)) and general indicators (aerobic mesophilic total viable count, aerobic mesophilic spore-forming bacteria, lactic acid bacteria, coliforms, *Enterobacteriaceae*, *Pseudomonas* spp., Sulfite-reducing anaerobes, yeasts/molds). In addition, the potential transfer of contaminants between substrates, larvae, and remaining frass was explored.

As a general trend, the systemic review and our experimental results show a high level of contamination of larvae on mesophilic aerobic bacteria, aerobic endospore bacteria, *Enterobacteriaceae*, and lactic acid bacteria. The main pathogenic bacteria found in samples included *Bacillus cereus group*, *Clostridium perfringens*, *Cronobacter* spp., *Escherichia coli*, and *Salmonella* spp., while *Campylobacter* spp., *Listeria monocytogenes*, and *Staphylococcus aureus* (coagulase-positive) were not detected, but the presence of *Staphylococcus aureus* was detected with a level below 10 cfu/g in a frass tested previously in one of the farms investigated.

These results are in line with data collected in our systematic review. Indeed, Osimani et al. [52] reported a high load of aerobic mesophilic total viable count, aerobic mesophilic spore-forming bacteria, and lactic acid bacteria in frass, with values up to 9.9, 7.6, and 8.2 log cfu g^−1^, respectively. More specifically, as illustrated in the article by Wynants et al. [32], substrate composition and intrinsic parameters have an impact on microbial composition. This variability in rearing substrates may explain the more heterogeneous distribution of substrates, particularly in terms of lactic acid bacteria (LAB) parameters, but this does not seem to have any impact on the microbial load of the larvae. LAB has also been found in high concentrations in other studies up to 7.4 log cfu g^−1^ [56]. The study by Luparelli et al. [57] highlighted the impact of fermentation on the molecular composition of biomasses, with an enrichment in monounsaturated and polyunsaturated fatty acids and essential amino acids, which could lead to antioxidant and antimicrobial activities.

Regarding the potential contamination by the main pathogenic bacteria, our study did not reveal the presence of *Bacillus cereus*, *Campylobacter* spp., *Listeria monocytogenes*, or *Salmonella* spp. in the samples of larvae reared on different substrates. All our larvae samples are negative for *Bacillus cereus* despite 17% to 100% positive samples for our substrate and frass samples, respectively. This is a surprising result because the presence of *B. cereus* in raw or processed larvae has already been observed [32,44,48,53]. This result can be explained by the limitations of our sample design, which does not allow for statistical coverage of sporadic or low-prevalence contaminations Hence, there is a need to improve detection methods; this pathogen has been found in samples of frass and substrates [48]. The Salmonella-positive frass sample does not appear to be an isolated case, as this pathogen has been found in the literature to be present in frass or raw larvae [32,44,46,51,52,53].

In the light of our data, it does not appear that substrates based on food at shelf life or containing meat are more contaminated than others are, and the larvae produced are of similar quality. A number of studies have shown that bacteria can be transferred from the substrate to the larvae [47], so it is essential to control the initial contamination of the substrate, but also the rearing method, and implement an effective inactivation process [44,49,50].

### 4.1. Microbiological Analysis of Hermetia illucens Larvae, Substrate, and Frass

#### 4.1.1. Levels of Microbial Indicators in Substrate, *Hermetia illucens* Larvae, and Frass

##### Substrate

Insect farming has emerged as a sustainable solution for producing protein-rich food and feed. A critical aspect of insect farming is the selection of substrates, which serve as the primary nutrient source for insects [58]. Due to their ability to thrive on a variety of organic substrates, *H. illucens* larvae show promising potential as an insect for both food and feed purposes [20,38,59,60]. In our study, we highlight differences in indicators quantities such as mesophilic aerobic bacteria and aerobic endospore bacteria in substrates. Additionally, concerning lactic acid bacteria, a wide variability was observed across substrates, ranging from less than 1 to 7.5 log cfu/g. Based on our statistical results, variations in contamination levels were observed across the classic, vegetable co-product, meat-based product, and food at shelf life substrates concerning bacterial endospores, lactic acid bacteria, and molds. Moreover, the contamination levels of these indicators in meat-based products differ from those observed in vegetable co-products and food at shelf life. The meat-based product substrate, made with a blend of vegetables supplemented with 1% meat, had contamination levels that differed from other substrates used for rearing *Hermetia illucens*. Furthermore, the levels of endospores found in the substrates suggest a need for careful substrate selection to mitigate contamination issues. Thus, depending on the substrate choice, there may be an impact on the microbiological quality of the larvae intended for feed and food consumption. Many studies tend to mainly focus on assessing how environmental conditions and diet affect the development and nutritional quality of *Hermetia illucens*, while frequently disregarding the potential influences of the microbiota present in these substrates on its growth performance and the final food product quality [58,60]. For example, Gold et al.’s research showed that deactivating the initial bacterial community in the rearing substrate could reduce rearing efficiency and uncover significant members of the microbiota that influence enhanced rearing performance [61]. Thus, the choice of breeding substrate for *Hermetia illucens* as well as the microbiome associated with the substrate would have strong impacts on the zootechnical performance of the insect and on the microbial quality of insect-based products (e.g., powder). Our study revealed significant variations in the levels of lactic acid bacteria depending on the rearing substrate types. It has been shown that substrates such as fermented food, and plant/grain-based substrates were dominated with LAB [32,62,63]. Certain lactic acid bacteria (LAB) strains are known to enhance the growth of insect and can be used for feed additives in farming industry to stimulate the growth of animals (poultry, pigs) [64,65,66]. Indeed, studies conducted by Somroo et al. and Mazza et al. have demonstrated that supplementing the substrate with *Lactobacillus buchneri* can enhance the rearing performance of black soldier fly larvae [66,67]. Given the variability in the presence of lactic acid bacteria depending on the substrates, it could be interesting to conduct a sampling campaign to isolate these microorganisms, which may possess anti-microbial potential or fermentation capabilities.

##### Frass

Several studies have explored comparable microbiological parameters in black soldier fly larvae (BSFL) frass [32,52,54,61]. The values depicted in Figure 3 align with the data on frass reported by Wynants et al. [32]. Their study encompassed various black soldier fly larvae rearing cycles conducted both at laboratory scale and in large-scale rearing facilities at different locations, each using other substrates and/or rearing conditions. The authors presented total viable counts for frass ranging from 8.5 to 10.2 log cfu/g across all rearing cycles. *Enterobacteriaceae* counts varied from <5.0 to 9.5 log cfu/g, while LAB counts ranged from <5.0 to 9.8 log cfu/g. Gold et al.’s [61] investigation into BSFL frass following a 12-day rearing cycle on canteen food waste and household food waste reported total viable counts of 8.6 to 10.0 log cfu/g, along with LAB counts of 7.5 to 8.1 log cfu/g. Osimani et al. [52] reported values similar to our study for BSFL frass reared until the prepupal stage on coffee silverskin with or without microalgae, with average values of 9.3 to 9.8 log cfu/g and 7.6 to 8.2 log cfu/g for total viable count and LAB, respectively. In addition, the same average counts of 3.7 to 5.0 log cfu/g were observed for *Enterobacteriaceae*. In another study by Van Looveren et al. [54], the average total viable count for BSFL frass, reared on a mixture of potato starch, a wheat and potato processing product, and protein kibbles, was 8.7 log cfu/g.

Discrepancies in rearing conditions, such as variations in the sampling moment within the rearing cycle, feed rates, and larval density, may differ across studies and have the potential to impact the microbiological counts of the frass [32,61]. In addition, Van Looveren et al. [55] suggested that the feeding method may influence the intrinsic parameters (water activity, pH, and moisture content) of the substrate/frass and thus the microbial count. More research is needed to determine the exact influence of the feeding system on the microbiota of frass.

##### Larvae

Insects naturally host diverse microbial communities, encompassing bacteria, yeasts, and fungi, crucial for their development, digestion, and overall well-being [68,69,70]. This study reveals that raw edible insects generally exhibit high levels of mesophilic aerobes, bacterial endospores, *Enterobacteriaceae*, lactic acid bacteria, psychotropic aerobes, fungi, and pathogenic species. Furthermore, our results demonstrate that there is no significant difference in microbiological levels between larvae and the frass. Interestingly, Kashiri et al. [49] revealed a high contamination load of total aerobic mesophilic bacteria (1.58 × 10^7^ cfu/g) and *Enterobacteriaceae* (1.15 × 10^6^ cfu/g) in the larvae. It is known that there is vertical transmission of microorganisms in *H. illucens* (from adults to eggs) [71].

On the other hand, multiple studies show that BSF larvae possess antimicrobial capacities and are able to reduce pathogenic fungi and bacteria such as *Salmonella*, *Staphylocccus aureus*, and *E. coli* in their substrate [38,39,59,72,73,74,75]. Furthermore, additional microorganisms within the rearing environment might hold promise for enhancing the insect’s growth performance and for leveraging antimicrobial peptides as alternatives to antibiotics in livestock farming [58,76,77,78,79,80].

#### 4.1.2. Major Pathogens Found in Substrate, *Hermetia illucens* Larvae, and Frass

Insects may also harbor undesirable microorganisms, including foodborne pathogens, posing potential risks to human health [44,81,82]. Our study based on culture-dependent methods highlights the presence of pathogens like *Bacillus cereus*, coliforms (including *E. coli β-glucuronidase-positive*, *Cronobacter*), and *Salmonella* spp. in frass, substrate, and larvae. Notably, *Staphylococcus aureus*, *Listeria monocytogenes*, and *Campylobacter* sp. were not detected. Interestingly, Kashiri et al. [49] revealed that the presence of pathogenic microorganisms varied: no *Listeria* spp. were found, but *Salmonella* (1.15 × 10^6^ cfu/g) and *E. coli* (7.08 × 10^5^ cfu/g) were detected in the larvae extract. While Grabowski et al. [48] found raw edible insects to be free of *Salmonellae*, *E. coli*, *L. monocytogenes*, and *Staphylococcus aureus*, they identified coagulase-negative staphylococci, *Enterobacteriaceae* (typically *Proteus* spp. and *Serratia liquefaciens*), pseudomonads, and fungi. Moreover, others authors reported the presence of *S. aureus* in the larvae and/or residue after rearing [32,53]. The substrate composition could drive the microbial communities by influencing the survival of pathogenic bacteria and could include specific microorganisms to limit pathogenic bacteria development [58].

### 4.2. Impact of Susbtrates on Hermetia illucens Comtamination

Our work suggests significant differences in contamination levels depending on the rearing substrate type as well as the farm. Therefore, the choice of substrate could influence the microbiological quality of the larvae. Different studies, using artificial contamination of the substrate, have shown that the contamination of insect substrate will contaminate larvae and thus represents a potential risk for human consumption [39,47]. There is also interest in investigating the potential of reduction in contamination by larvae as in the study by Erickson et al. [39], in which BSF larvae were reared with chicken manure and an improved inactivation of *Salmonella* and *E. coli* O157:H7 in alkaline chicken manure was found. The BSF was also found to produce antimicrobial peptides with a large diversity [83,84,85,86].

### 4.3. Microbial Criteria of Control

Given the elevated initial bacterial load, additional information is needed on processing techniques to guarantee the quality of the final insect-derived products, especially for large-scale production. Comparing the results with the hygiene criteria for edible insects proposed by Belgium [87] and the Netherlands (NVWA, Netherlands Food and Consumer Product Safety Authority, 2014), it was observed that powdered insect products did not meet several bacterial count limits, even in the absence of classical food pathogens [50]. Consequently, it is recommended that edible insects be consistently consumed after an additional heating step, as instructed by the manufacturer, until advancements in drying techniques can guarantee lower bacterial counts. This underscores the need for exploring alternative processing methods to effectively regulate contamination levels in insects. In order to control pathogenic bacteria, inactivation studies of natural contaminating microorganisms were conducted [45,49,50,51]. A microbiological characterization on *Hermetia illucens* larvae as well as an inactivation study of natural contaminating microorganisms and inoculated *E. coli* O157:H7 by using High Hydrostatic Pressure (HHP) was carried out by Kashiri et al. [49]. HHP demonstrated effectiveness in combating naturally occurring yeasts and molds, resulting in reductions exceeding five log cycles at 400 MPa for any of the considered durations (ranging from 2.5 to 7 min). However, it resulted in only a modest reduction in the overall microbial load. The level of inactivation of larvae inoculated with *E. coli* O157:H7 showed variability. The study of Campbell et al. [45] presented the impact of thermal (90 °C for 10/15 min) and high-pressure processing (HPP; 400/600 MPa for 1.5/10 min) treatments on the microbial levels on *Hermetia illucens* larvae. Nyangena et al. [51] examined the effects of traditional processing techniques (boiling, toasting, solar-drying, oven-drying) on microbiological quality of *Hermetia illucens* larvae. Boiling and toasting were effective in reducing aerobic mesophilic bacterial populations, decreasing *Staphylococcus aureus*, and eliminating yeasts, molds, Lac+ enteric bacteria, and *Salmonella*. Oven-drying alone had a marginal impact on bacterial populations and yeast/mold levels, while solar-drying alone did not affect these parameters. However, oven-drying boiled or toasted products increased aerobic mesophilic bacteria counts, yet the products continued to be free of Lac+ enteric bacteria and *Salmonella*. In another study conducted by Larouche et al. [50], the objective was to enhance the efficacy of larval extermination by evaluating the impact of 10 different methods on microbiological qualities. These methods included blanching (B = 40 s), desiccation (D = 60 °C, 30 min), freezing (F20 = −20 °C, 1 h; F40 = −40 °C, 1 h; N = liquid nitrogen, 40 s), high hydrostatic pressure (HHP = 3 min, 600 MPa), grinding (G = 2 min), and asphyxiation (CO_2_ = 120 h; N_2_ = 144 h; vacuum conditioning, V = 120 h). The findings revealed that certain methods had an impact on intrinsic parameters such as pH, total moisture, and ash contents. Blanching emerged as the most effective strategy for minimizing microbial contamination. Future research should be undertaken to mimic more realistically rearing facility substrate inactivation technologies (e.g., pasteurization) without altering the nutritional quality of the insect-based product.

### 4.4. Limitis of Cultural Methods and Potential of Combining with Sequencing Methods

The presence of these high bacterial counts and (opportunistic) pathogenic bacteria emphasizes the need for a deeper understanding of microbial communities in the insect matrix and its environment, necessitating identification of pathogenic bacteria at the species level. Traditional culture-based microbiology methods have continued to be the most effective means of identifying, quantifying, and selecting dominant microorganisms in various food sources, including insects [88]. However, these techniques exhibit limitations in terms of their detection thresholds and the accurate assessment of “real” diversity (e.g., 0.1 to 10% of cultivable bacteria within a total microbial community, depending on the specific biotope under investigation). Furthermore, conventional culture-based methodologies face challenges in isolating pathogenic bacteria due to overgrowth by concomitant microflora, compounded by the lack of strictly selective mediums for insect matrices. Consequently, new strategies have been developed to highlight taxonomic groups that remain elusive when employing conventional methods [89]. Significant strides have been made in characterizing insect-associated microbial communities through advanced molecular techniques like high-throughput DNA sequencing [90,91,92,93,94]. Molecular techniques, such as metagenomics, metagenetics, transcriptomics, proteomics, and functional genomics, provide deeper insights into bacterial physiology and offer non-culture-based identification methods [95,96]. For instance, 16S rRNA gene sequencing found *Campylobacter* bacteria to be prevalent in the gut of unprocessed black soldier flies [97]. Other studies have also been able to define the microbiological quality of post-processed foods or even the effect of insect processing techniques on the food [51,98,99].

This preliminary approach opens avenues for controlling transformation conditions to enhance food safety, addressing concerns like antibiotic resistance facilitated by high insect densities [100]. Future research should focus on identifying specific strains unique to insects or commonly associated with foodborne illnesses, providing a nuanced understanding of microbial strains and their implications for food safety within the broader context of insect-associated risks.

## 5. Conclusions

*Hermetia illucens* larvae are increasingly being considered as an alternative source of protein for animal and human food. Managing microbial risk is necessary to ensure animal and human health. Insect larvae are interesting because they develop quickly on numerous substrates, but they are also potential vectors of biological hazards, since they are used in their entirety with their digestive tract and are difficult to separate hygienically from their substrate and frass.

The present study aimed to evaluate the microbiological quality of BSF larvae, performing a systematically review of the literature and conducting microbial analysis in three farms using four different rearing substrates including traditional recipes (cereals, fruits, vegetables), vegetable agri-food co-products, former foodstuffs arriving at shelf life and a meat-based option. Available data are limited, with, to date, only 13 studies reporting BSF microbiological analysis data. These data, combined with our experimental plan, confirm the high level of contamination of larvae for the main indicators and the potential contamination in pathogenic bacteria including *Bacillus cereus* group, *Campylobacter* spp., *Clostridium perfringens*, *Cronobacter* spp., *Escherichia coli*, *Salmonella* spp., and *Staphylococcus aureus.* Larvae appear in the literature and in our experiments as vectors of contamination transmitted by the substrate. It is therefore important to ensure its quality. None of the four substrates investigated (including cereals, fruits, vegetables, vegetable co-products, food at shelf life, or meat) were free of pathogenic bacteria. Results must be considered with regard to the limited numbers of samples analyzed (three repetitions involving three farms with their substrates) as a heterogeneity was observed which can be due to the sampling plan being limited in case of sporadic contamination in substrate and larvae, leading to difficult identification of contamination. The most contaminated substrate observed was vegetable co-products including positives in *Clostridium perfringens*, *Cronobacter* spp., *Escherichia coli*, and *Bacillus cereus* leading to larvae also contaminated in the three bacteria listed. The main trends were the substrate contaminated with *Bacillus cereus* and larvae contaminated with *Escherichia coli.* On top of that, the classic substrate was also contaminated with *Clostridium perfringens* and *Cronobacter* spp.

Considering the microbial contamination of *Hermetita illucens* raw larvae, the main recommendation if to implementation of a critical control point (CCP) is essential at the manufacturing step in the HACCP system; however, it can only reduce vegetative forms in the case of a boiling step [101], and will not efficiently reduce spore-forming bacteria. This reinforces the need to control pathogenic bacteria at every step of the farm-to-fork chain. It must be noted that the drying step is less efficient then a boiling step [101]. The efficiency of this inactivation step can be optimized using predictive microbiology tools [102]. Thus, the monitoring of spore-forming bacteria in substrates seems to be necessary in view of their high frequency of contamination in all substrates, particularly *Bacillus cereus* and *Clostridium perfringens*.

Methods of control of larvae using culture-based approaches are challenged by numerous and high levels and contaminations and can be reinforced by the use of “omics” such as metagenomic for a better understanding of the microbial composition, functional potential, and interactions within the larval ecosystem. These approaches can help identify specific microbial species, their genetic traits, and potential interactions, providing valuable insights into the dynamics of microbial communities associated with different rearing substrates. These investigations can contribute to the development of targeted strategies for optimizing larval rearing conditions, improving microbiological safety, enhancing the nutritional value, and establishing efficient and sustainable production systems for *Hermetia illucens* larvae, ensuring their safety and quality for animal feed and human consumption.

## Figures and Tables

**Figure 1 foods-13-01587-f001:**
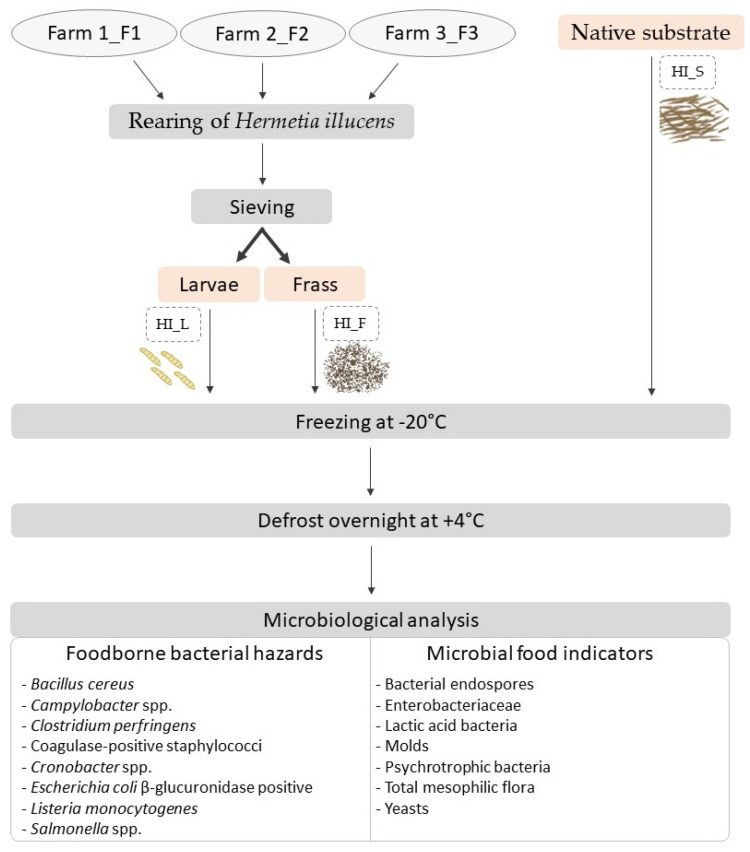
Experimental design of microbiological analysis (gray: main steps, orange: samples collected, and dotted line: sample codes).

**Figure 2 foods-13-01587-f002:**
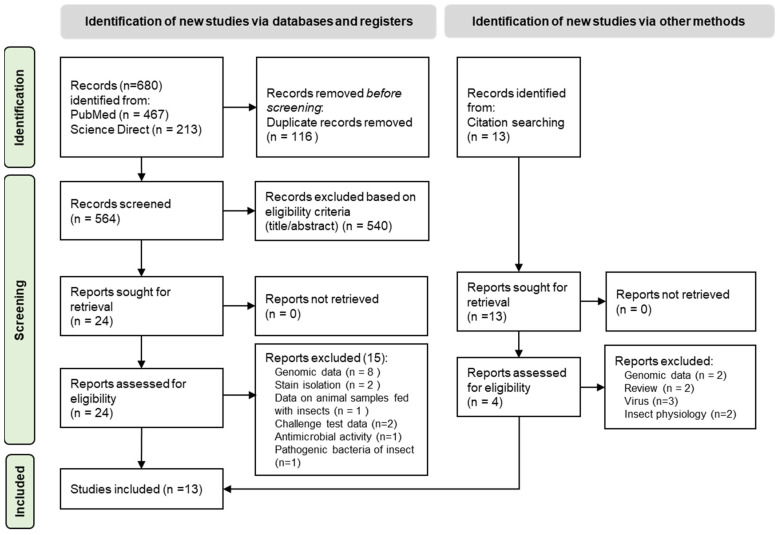
PRISMA systemic review on black soldier fly microbial quality.

**Figure 3 foods-13-01587-f003:**
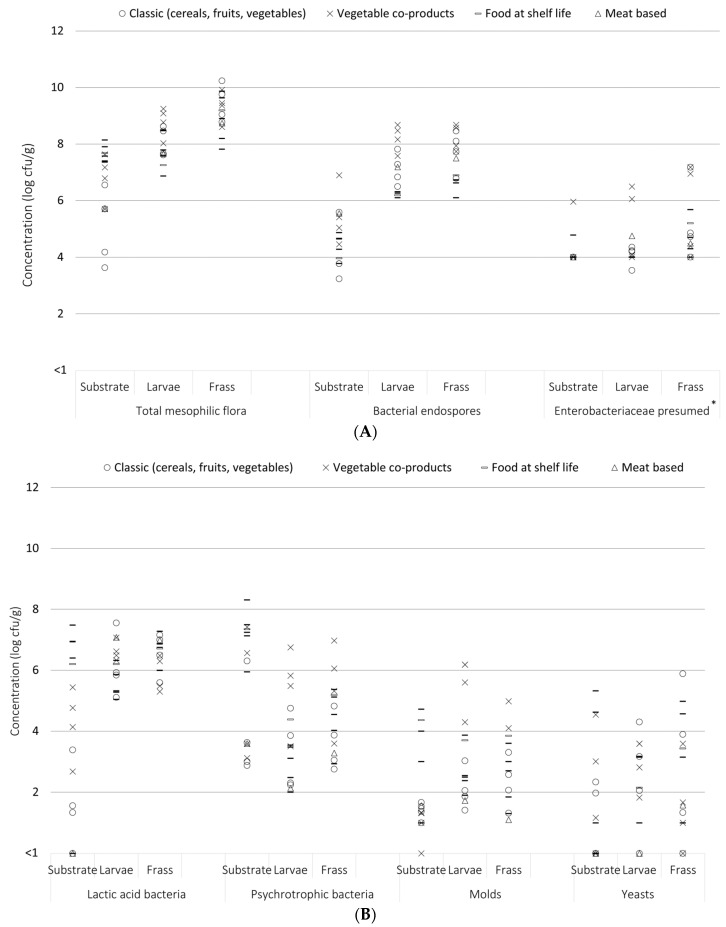
Summary of the results of the enumeration of the main indicators: mesophilic aerobic bacteria, aerobic endospore bacteria, and Enterobacteriacea (**A**), and lactic acid bacteria, psychrotrophic bacteria, molds, and yeasts (**B**). Values are expressed in log cfu/g for the native substrate, larvae, and frass rearing on various substrates (classic, vegetable co-products, food at shelf life, or meat). * Limit of quantification for Enterobacteriaceae presumed <4 to >7 log cfu/g.

**Table 1 foods-13-01587-t001:** Substrates tested and number of batches produced according to the breeding of black soldier fly larvae.

Rearing Farm	Substrate Category	Substrate Recipes	Replicate
Farm 1 F1	Classic (cereals, fruits, vegetables)	Apple pomace and wheat bran	3 batches
	Classic (cereals, fruits, vegetables)	Fruit peel, fruit, vegetable, wheat bran, brewer’s grains, chestnut chips	3 batches
	Food at shelf life	Sandwich at shelf life	1 batch chicken, 1 batch tuna, 1 batch ham
	Food at shelf life	Meat-based dish prepared at shelf life	1 batch «Andalusian», 1 batch «Italian», 1 batch «Vendean»
	Meat-based	Fruit peel, fruit, vegetable, wheat bran, brewer’s grains, chestnut chips with 1% beef	3 batches
Farm 2 F2	Classic (cereals, fruits, vegetables)	Bananas, variable, wheat bran	3 batches
Farm 3 F3	Classic (cereals, fruits, vegetables)	Organic poultry food *	3 batches
	Vegetable co-products	Celery pomace	3 batches
	Vegetable co-products	Carrot pomace	3 batches
	Vegetable co-products	Onion pomace	3 batches
	Vegetable co-products	Wheat Silo	3 batches

* The recipe includes wheat, corn, soybean expeller, calcium carbonate, barley, wheat bran, corn, corn gluten, sunflower expeller, alfalfa, dicalcium phosphate, and paprika extract.

**Table 2 foods-13-01587-t002:** Microbiological analysis methods for all the screened bacterial indicators or pathogenic agents.

Parameters	Standards
*Bacillus cereus* (30 °C)	BKR 23/06-02/10 *
*Campylobacter* spp.	NF EN ISO 10272-1/A1
*Clostridium perfringens*	NF EN ISO 7937
Coagulase-positive staphylococci (37 °C)	NF EN ISO 6888-2
*Cronobacter* spp.	NF EN ISO 22964
Enterobacteriaceae (37 °C)	NF V08-054
*Escherichia coli* β-glucuronidase-positive	Adapted from NF ISO 16649-2 (TBX, 21 h ± 3 h) NF ISO 16649-2
Lactic acid bacteria (30 °C)	NF ISO 15214
*Listeria monocytogenes*	AES 10/03-09/00/BKR 23/02-11/02 **
Molds (on products at a_w_ < 0.96)	Internal methodology (DG18, 120 h ± 3 h) [Gélose GGC 25 °C–120H+/−3H]
*Salmonella* spp.	BRD 07/11-12/05 BKR 23/07-10/11 ***
Yeasts/molds (on products at a_w_ < 0.96)	NF V 08-036
Yeasts (on products at a_w_ < 0.96)	Internal methodology (DG18, 120 h ± 3 h)

* refer to COMPASS® *Bacillus cereus* Agar: alternative analysis method validated by AFNOR (Association Française de Normalisation) Certification for the enumeration of *Bacillus cereus*; ** refer to ALOA® ONE DAY: alternative analysis method validated by AFNOR Certification for the detection of *Listeria* spp. and *Listeria monocytogenes*; *** refer to RAPID’Salmonella: alternative analysis method validated by AFNOR Certification for the detection of *Salmonella* spp.

**Table 3 foods-13-01587-t003:** Synthesis of studies included from the scientific literature for data extraction.

Samples and BSF Larvae Forms Analyzed																		
Author	Raw Larvae (Unprocessed)	Processed Larvae	Substrates	Frass	Asphyxia	Boiled	Desiccation	Dried (Unspecified)	Fat	Freeze-Dried	Frozen Slaughter	High Pressure	Microwave-Dried	Oven-Dried	Powder	Solar-Dried	Toasted	Sample’s Origin
Bessa et al., 2021 [44]	X	X				X					X							South Africa
Campbell et al., 2020 [45]	X	X				X					X	X						Ireland
De smet et al., 2021 [46]	X		X	X														Belgium
Gorrens et al., 2021a [47]	X		X	X														Belgium
Grabowski et al., 2017 [48]		X						X	X						X			Germany or the Netherlands or Europe or Asia
Kashiri et al., 2018 [49]		X						X			X	X						Spain
Larouche et al., 2019 [50]	X	X			X	X	X				X	X						Canada
Nyangena et al., 2020 [51]	X	X				X								X		X	X	Kenya
Osimani et al., 2021 [52]	X		X	X														Italy
Raimondi et al., 2020 [53]	X										X							Italy
Van Looveren et al. 2022a [54]				X														Belgium
Van Looveren et al. 2022b [55]	X	X	X	X				X										Insect rearing company
Wynants et al., 2018 [32]	X		X	X														Belgium, The Netherlands, Switzerland

**Table 4 foods-13-01587-t004:** Summary of micro-organisms levels found during the breeding of *Hermetia illucens* (substrates, larvae, frass) with or without processing.

Microbiological Parameter	Levels Reported of *Hermetia illucens* Larvae, Substrate, or Frass (log cfu/g)											
	Substrate		Frass		Larvae							
					Raw ^1^		Heat-Treated ^2^		Dried ^3^		Fat	
	min	max	min	max	min	max	min	max	min	max	min	max
Aerobic mesophilic total viable count	2.6	>11.5	3.7	12.4	5.5	>10.7	2.1	8.1	1.6	7.8	5.4 *	
Aerobic mesophilic spore-forming bacteria	1.1	6.3	3.2	7.6	2.4	7.9	/	/	2.9	3.9	/	/
Lactic Acid Bacteria	<1.0	9.0	<5.0	9.8	3.5	9.9	<2.0	6.7	7.8		/	/
*Aspergillus* spp.	/	/	/	/	/	/	/	/	/	/	+	
*Bacillus cereus group*	<2.0		n.d.	<3.7	n.d.	<3.8	1.7	2.0	n.d.	+	+	4.6
*Campylobacter* spp.	<2.0		/	/	3.2	4.7	/	/	/	/	/	/
*Clostridium perfringens*	<1.0	3	n.d.	2.2	0.8	1.6	/	/	<1.0		/	/
Coliforms	/	/	/	/	4.5	7.6	/	/	/	/	/	/
*Enterobacteriaceae*	<1.0	4.6	<1.0	>9.6	2.9	9.7	0.0	6.1	0.0	8.1	0.0	
*Escherichia coli*	/	/	/	/	4.5		1.3	1.5	5.9		/	/
*Listeria monocytogenes*	n.d.		n.d.		n.d.		n.d.		n.d.		/	/
*Listeria* spp.	2.6		/	/	4.8	7.0	5.2	5.5	5.2		/	/
*Pseudomonas* spp.	/	/	/	/	5.6	7.8	<2.1		4.8		/	/
*Salmonella* spp.	n.d.	8.1	n.d.	9.5	n.d.	<5.9	n.d.		nd	6.1	/	/
*Staphylococcus aureus (coagulase-positive)*	<2.0	6.6	n.d.	7.5	n.d.	8.4	2.5		0.9	3.0	4.4	
Sulfite-reducing anaerobes	/	/	<1.0		8.4	9.9	4.8	6.1	7.9		/	/
Yeasts/molds	<2.0	7.7	3.6	7.8	0.7	7.6	0.0	7.8	0.0	6.8	+	2.9

^1^ With or without effect of rinsing and/or storage, larvae slaughtered by asphyxiation. ^2^ Boiled, high hydrostatic pressures, high-pressure. ^3^ Toasted, oven-dried, solar-dried, toasted and oven-dried, toasted and solar-dried, boiled and solar-dried, desiccation. * Only one value available, no min or max, or same value for all samples, / no value reported, n.d. not detected, and + present. See Appendix A for more details of limit quantification.

**Table 5 foods-13-01587-t005:** Microbiological contamination levels (log CFU/g) per rearing farm. Data are the mean of replicates ± standard deviation. Within each row, means followed by different letters (a, b) are significantly different (*p* < 0.05).

Microbiological Parameter	Farm 1	Farm 2	Farm 3
Bacterial endospores	6.3 ± 1.4 ^a^	6.6 ± 2 ^ab^	7.1 ± 1.7 ^b^
Enterobacteriaceae	4.5 ± 0.9 ^a^	4.2 ± 1 ^a^	4.7 ± 1.2 ^a^
Lactic acid bacteria	5.3 ± 2.4 ^a^	4.8 ± 1.1 ^a^	5.5 ± 1.7 ^a^
Molds	2.2 ± 1.2 ^a^	1.8 ± 1.3 ^ab^	3.2 ± 1.9 ^b^
Psychotrophic bacteria	4.0 ± 1.6 ^a^	4.6 ± 1.3 ^ab^	5.0 ± 1.7 ^b^
Total mesophilic flora	7.7 ± 1.6 ^a^	8.2 ± 1.5 ^a^	8.1 ± 1.6 ^a^
Yeasts	2.2 ± 1.7 ^a^	2.6 ± 1.4 ^a^	2.2 ± 1.8 ^a^

**Table 6 foods-13-01587-t006:** Microbiological contamination levels (log CFU/g) per type of sample. Data are the mean of replicates ± standard deviation. Within each row, means followed by different letters (a, b, c) are significantly different (*p* < 0.05).

Microbiological Parameter	Frass	Larvae	Substrate
Bacterial endospores	7.8 ± 0.8 ^a^	7.4 ± 1.0 ^a^	4.8 ± 1.1 ^b^
Enterobacteriaceae	5.1 ± 1.3 ^a^	4.4 ± 0.9 ^ab^	4.2 ± 0.7 ^b^
Lactic acid bacteria	6.4 ± 0.7 ^a^	6.3 ± 1.1 ^a^	3.4 ± 2.2 ^b^
Molds	2.9 ± 1.7 ^a^	3.2 ± 1.7 ^a^	1.7 ± 1.1 ^b^
Psychotrophic bacteria	4.4 ± 1.4 ^a^	4.1 ± 1.7 ^b^	5.0 ± 1.9 ^c^
Total mesophilic flora	9.3 ± 0.7 ^a^	8.2 ± 0.7 ^b^	6.2 ± 1.4 ^c^
Yeasts	2.3 ± 1.9 ^a^	2.4 ± 1.7 ^b^	2.0 ± 1.5 ^c^

**Table 7 foods-13-01587-t007:** Microbiological contamination levels (log CFU/g) per category of substrate. Data are the mean of replicates ± standard deviation. Within each row, means followed by different letters (a, b, c and d) are significantly different (*p* < 0.05).

Microbiological Parameter	Classic	Vegetable Co-Products	Food at Shelf Life	Meat Based
Bacterial endospores	6.4 ± 1.8 ^a^	5.8 ± 1.1 ^b^	6.7 ± 0.9 ^ca^	7.4 ± 1.5 ^abc^
Enterobacteriaceae	4.4 ± 1.1 ^a^	4.3 ± 0.5 ^a^	4.4 ± 0.4 ^a^	4.8 ± 1.3 ^a^
Lactic acid bacteria	4.8 ± 2.4 ^a^	6.0 ± 1.4 ^a^	5.0 ± 3.0 ^a^	5.6 ± 1.3 ^a^
Molds	1.9 ± 1.1 ^a^	3.0 ± 1.0 ^cd^	1.5 ± 0.5 ^bc^	3.5 ± 2.0 ^da^
Psychotrophic bacteria	3.7 ± 1.2 ^ab^	5.0 ± 1.9 ^cd^	3.0 ± 0.8 ^ad^	3.5 ± 2.0 ^da^
Total mesophilic flora	7.6 ± 2.1 ^a^	8.1 ± 0.8 ^a^	7.4 ± 1.3 ^a^	3.5 ± 2.0 ^da^
Yeasts	2.3 ± 1.7 ^a^	2.4 ± 1.6 ^a^	1.2 ± 0.4 ^a^	3.5 ± 2.0 ^da^

**Table 8 foods-13-01587-t008:** Synthesis of major bacterial pathogen detection for the three farms included in the experimental plan.

	Substrate (n = 33)			Larvae (n = 34)			Frass (n = 36)		
	F1 (n = 15)	F2 (n = 3)	F3 (n = 15)	F1 (n = 15)	F2 (n = 3)	F3 (n = 16)	F1 (n = 15)	F2 (n = 4)	F3 (n = 17)
*Campylobacter* spp.	0%	0%	0%	0%	0%	0%	0%	0%	0%
*Clostridium perfringens* presumed	7%	0% *	7%	0% *	0% *	69%	0% *	25%	6%
Coagulase-positive staphylococci	0% *	0% *	0% *	0% *	0% *	0% *	0% *	0% *	0% *
*Cronobacter* spp.	13%	0%	20%	0%	0%	0%	0%	0%	35%
*Escherichia coli* β-glucuronidase-positive	7%	0% *	20%	87%	33%	87.5%	87%	75%	59%
*Listeria monocytogenes*	0%	0%	0%	0%	0%	0%	0%	0%	0%
Presumptive *Bacillus cereus*	87%	67%	53%	0% *	0% *	0% *	47%	100%	65%
*Salmonella* spp.	0%	0%	0%	0%	0%	0%	0%	0%	6%

* All samples had results <1 log cfu/g, below the limit of quantification.

**Table 9 foods-13-01587-t009:** Synthesis of major bacterial pathogen detection for the four types of substrates.

	Substrate (n = 33)				Larvae (n = 34)				Frass (n = 36)			
	Classic (Cereals, Fruits, Vegetables) (n = 10)	Vegetable Co-Products (n = 14)	Food at Shelf Life (n = 6)	Meat (n = 3)	Classic (n = 10)	Vegetable Co-Products (n = 15)	Food at Shelf Life (n = 6)	Meat (n = 3)	Classic (n = 10)	Vegetable Co-Products (n = 17)	Food at Shelf Life (n = 6)	Meat (n = 3)
*Campylobacter* spp.	0%	0%	0%	0%	0%	0%	0%	0%	0%	0%	0%	0%
*Clostridium perfringens* presumed	10%	7%	0% *	0% *	0% *	73%	0% *	0% *	0% *	12%	0% *	0% *
Coagulase-positive staphylococci	0% *	0% *	0% *	0% *	0% *	0% *	0% *	0% *	0% *	0% *	0% *	0% *
*Cronobacter* spp.	10%	14%	33%	0%	0%	20%	0%	0%	0%	35%	0%	0%
*Escherichia coli* β-glucuronidase-positive	0% *	21%	17%	0% *	100%	53%	83%	100%	80%	65%	67%	100%
*Listeria monocytogenes*	0%	0%	0%	0%	0%	0%	0%	0%	0%	0%	0%	0%
Presumptive *Bacillus cereus*	100%	43%	67%	100%	0% *	0% *	0% *	0% *	60%	71%	17%	67%
*Salmonella* spp.	0%	0%	0%	0%	0%	0%	0%	0%	0%	6%	0%	0%

* All samples had results <1 log cfu/g, below the limit of quantification.

## Data Availability

The original contributions presented in the study are included in the article/Appendix A, further inquiries can be directed to the corresponding author.

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
