# Peer review of "Evaluation of the Microbial Quality of Hermetia illucens Larvae for Animal Feed and Human Consumption: Study of Different Type of Rearing Substrates"

_foods, 2024, doi:10.3390/foods13101587_

Round 1
Reviewer 1 Report
Comments and Suggestions for Authors
I have reviewed your manuscript titled "Evaluation of the Microbial Quality of Hermetia illucens Larvae for Animal Feed and Human Consumption: Study of Different Types of Rearing Substrates." The study addresses an important aspect of insect farming, specifically the microbial safety of Black Soldier Fly (BSF) larvae, which is a pertinent topic given the increasing interest in insects as a sustainable protein source. However, there are several areas that require improvement to strengthen the manuscript.
1. Limited sample size of the article: LINE 140, “2.2. Experimental design and sample collection”, 102 sample results from three farms were collected in the article, which may limit the generality and applicability of the results. To enhance the reliability of the study, it is recommended that the authors expand the sample sources in future work and consider more rigorous statistical methods for data analysis.
2. The way of data presentation and statistical analysis is not intuitive: although the article provides a large amount of data, the way of data presentation is relatively simple and lacks the assistance of charts and graphs. Authors are advised to use graphs to show key results such as the effects of different feeding substrates on microbial communities and the effects of different treatments on microbial reduction. In addition, appropriate statistical analyses are recommended to verify the significance and relevance of the results.
3. The finiteness of the discussion section: LINE 489, “5. Conclusions”, The conclusion section should clearly summarize the main findings of the study, and provide specific recommendations and future research directions. It is recommended that the authors clearly indicate the limitations of the study in the conclusion and suggest how to improve the microbial safety of insect products by improving rearing management and processing techniques.
In conclusion, the manuscript presents an interesting study with valuable findings. However, with the above-mentioned improvements, the manuscript could make a more significant contribution to the understanding of microbial quality in BSF larvae and its implications for the safety of insect-derived products.
Comments on the Quality of English Language
I suggest the Quality of English Language should be improved.
Author Response
We thank the editor and the two reviewers for the relevant suggestions and we acknowledge the time and effort on this article. We appreciate these two constructive feedbacks that helped to improve the quality and validity of our manuscript. You will find below a point-by-point answer and changes are included in yellow in the manuscript file.
We have copied below all Reviewers comments and our answers are written in bleu.
I have reviewed your manuscript titled "Evaluation of the Microbial Quality of Hermetia illucens Larvae for Animal Feed and Human Consumption: Study of Different Types of Rearing Substrates." The study addresses an important aspect of insect farming, specifically the microbial safety of Black Soldier Fly (BSF) larvae, which is a pertinent topic given the increasing interest in insects as a sustainable protein source. However, there are several areas that require improvement to strengthen the manuscript.
- Limited sample size of the article: LINE 140, “2.2. Experimental design and sample collection”, 102 sample results from three farms were collected in the article, which may limit the generality and applicability of the results. To enhance the reliability of the study, it is recommended that the authors expand the sample sources in future work and consider more rigorous statistical methods for data analysis.
We acknowledge the remark about the limited sample size in our paper. Collecting 102 samples from the three farms may indeed restrict the generalizability of the results but our objective was to explore the substrate diversity in terms of microbial quality. In addition, the significant cost associated with analysis was a limitation that we could not overcome. In future research we will use any opportunity to increase this collection and will try to include a broader range of farms to better represent the diversity of agricultural practices and environmental conditions.
Furthermore, to strengthen the robustness of our findings, we have incorporated additional statistical analyses in our study. See description in next comment.
- The way of data presentation and statistical analysis is not intuitive: although the article provides a large amount of data, the way of data presentation is relatively simple and lacks the assistance of charts and graphs. Authors are advised to use graphs to show key results such as the effects of different feeding substrates on microbial communities and the effects of different treatments on microbial reduction. In addition, appropriate statistical analyses are recommended to verify the significance and relevance of the results.
We agree with your observation and appreciate your suggestions for improving the data presentation in our article. We have taken your recommendations into account and have included additional statistical analysis to verify the significance and relevance of the results. We hope these modifications will meet your expectations and contribute to a better understanding of the results of our study. Thank you again for your constructive feedback.
Changes in the article in line with this comment are:
- Materials and methods section 2.5 (Lines 192-198)
- Tables 5, 6 and 7 including obtained results (Lines 246-256)
- Description of results in Lines 277-300 and 394-404
- The finiteness of the discussion section: LINE 489, “5. Conclusions”, The conclusion section should clearly summarize the main findings of the study, and provide specific recommendations and future research directions. It is recommended that the authors clearly indicate the limitations of the study in the conclusion and suggest how to improve the microbial safety of insect products by improving rearing management and processing techniques.
Thank you for this recommendation, we have included the objective of the study in the conclusion, see Lines 575-579. The summary of main findings can be seen in Lines 579-596. Recommendations are in Lines 597-606. Limitations are mentioned in Lines 587-591 and 607-617.
In conclusion, the manuscript presents an interesting study with valuable findings. However, with the above-mentioned improvements, the manuscript could make a more significant contribution to the understanding of microbial quality in BSF larvae and its implications for the safety of insect-derived products.
Thank you very much for your valuable feedback and help to improve our manuscript.
Comments on the Quality of English Language - I suggest the Quality of English Language should be improved.
We have tried to improve the text and had it checked by a more qualified but non-native speakers, who indicated that this level was sufficient for a scientific article. We hope that it will be adequate.

Reviewer 2 Report
Comments and Suggestions for Authors
The study appears to be meticulous and relevant to the field of food safety and entomophagy, addressing an emerging issue in sustainable protein production. The methodology used, including both systematic review and direct experimentation on farms, provides a solid basis for the conclusions drawn. The identification of specific pathogens and the absence of others is essential for the development of safe management practices in the breeding and processing of insects for consumption. Furthermore, the manuscript provides valuable information on the impact of different substrates on the microbiological quality of larvae, which is crucial for the regulation and implementation of safer breeding practices.
The study focuses on the microbiological quality of larvae and substrates, but could benefit from a more complete evaluation of management practices on farms, including hygiene, cross-contamination control, and risk mitigation strategies.
Could the authors explain the selection process for the four rearing substrates used in the study?
Were other potential substrates considered that could influence the microbiota of H. illucens larvae differently?
How do the authors ensure that the three farms selected for the study adequately represent the diversity of husbandry practices and environmental conditions in Hermetia illucens production?
Author Response
We thank the editor and the two reviewers for the relevant suggestions and we acknowledge the time and effort on this article. We appreciate these two constructive feedbacks that helped to improve the quality and validity of our manuscript. You will find below a point-by-point answer and changes are included in yellow in the manuscript file.
We have copied below all Reviewers comments and our answers are written in bleu.
The study appears to be meticulous and relevant to the field of food safety and entomophagy, addressing an emerging issue in sustainable protein production. The methodology used, including both systematic review and direct experimentation on farms, provides a solid basis for the conclusions drawn. The identification of specific pathogens and the absence of others is essential for the development of safe management practices in the breeding and processing of insects for consumption. Furthermore, the manuscript provides valuable information on the impact of different substrates on the microbiological quality of larvae, which is crucial for the regulation and implementation of safer breeding practices.
Thank you, we really appreciate the comment.
The study focuses on the microbiological quality of larvae and substrates, but could benefit from a more complete evaluation of management practices on farms, including hygiene, cross-contamination control, and risk mitigation strategies.
These topics are indeed relevant regarding the safety of larvae but the selected industries have already good hygienic practices in place and are monitored following the French national surveillance plan through inspections actions. Thus, we decided to focus on larvae quality and their substrates.
Could the authors explain the selection process for the four rearing substrates used in the study? Were other potential substrates considered that could influence the microbiota of H. illucens larvae differently?
The selection of substrates aimed to include both the traditional substrate commonly used in Hermetia illucens rearing, as well as alternative substrates aligned with waste reduction strategies. The traditional substrate chosen was the one typically used by each participating farm. The alternatives included by-products from the agri-food industry, products at expiration date from supermarkets, and grounded beef during their shelf-life. These substrates were chosen to reflect a variety of potential feed sources that could influence the contamination of H. illucens larvae differently, aligning with the objectives of our study to explore sustainable and diverse feeding options for larval development.
This was added in Lines 156-159.
How do the authors ensure that the three farms selected for the study adequately represent the diversity of husbandry practices and environmental conditions in Hermetia illucens production?
The three farms selected for the study are located in different regions of France, aiming to capture a diverse range of husbandry practices and environmental conditions in Hermetia illucens production. In addition, less than ten farms were identified in total, and all of them were contacted for potential participation in the study. Therefore, the selection of these three farms was based on the availability and willingness of farm owners to participate rather than a comprehensive sampling strategy.

Round 2
Reviewer 1 Report
Comments and Suggestions for Authors
I have carefully checked all the reply and no further question. I agree with this manuscript chould be published on the journal.